# Quantum Diffusion Model for Quark and Gluon Jet Generation

**Mariia Baidachna***
University of Glasgow

**Rey Guadarrama**
Benemérita Universidad Autónoma de Puebla

**Gopal Ramesh Dahale**
EPFL

**Tom Magorsch**
Technische Universität Dortmund

**Isabel Pedraza**
CERN

**Konstantin T. Matchev**
University of Florida

**Katia Matcheva**
University of Florida

**Kyoungchul Kong**
University of Kansas

**Sergei Gleyzer**
University of Alabama

## Abstract

Diffusion models have demonstrated remarkable success in image generation, but they are computationally intensive and time-consuming to train. In this paper, we introduce a novel diffusion model that benefits from quantum computing techniques in order to mitigate computational challenges and enhance generative performance within high energy physics data. The fully quantum diffusion model replaces Gaussian noise with random unitary matrices in the forward process and incorporates a variational quantum circuit within the U-Net in the denoising architecture. We run evaluations on the structurally complex quark and gluon jets dataset from the Large Hadron Collider. The results demonstrate that the fully quantum and hybrid models are competitive with a similar classical model for jet generation, highlighting the potential of using quantum techniques for machine learning problems.

## 1 Introduction

Denoising diffusion models (DDMs) have revolutionized the field of generative artificial intelligence (GenAI) by demonstrating their ability to generate high-quality images [6]. They overcome the drawbacks of generative adversarial networks (GANs), which are prone to mode collapse, becoming a new state-of-the-art architecture for image generation [11, 25]. Consequently, DDMs have been applied in many generative tasks for science from molecular biology to medical image synthesis to gravitational lensing [26, 2, 27, 18, 19].

Despite their successes, DDMs face significant challenges concerning the extensive computational resources required for training [24, 16]. New compute paradigms must be employed in order to overcome the computational bottleneck. Quantum machine learning (QML) offers a promising solution [21]. By cleverly incorporating quantum components into classical algorithms, quantum computers can efficiently solve problems that are difficult for classical computers with accelerated computations [8]. This paradigm shift has the potential to surpass current limitations and unlock the full potential of DDMs.

---

*Corresponding author: 2828197b@student.gla.ac.uk

38th Conference on Neural Information Processing Systems (NeurIPS 2024).

## 2 Background

### 2.1 Denoising Diffusion Model

There have been multiple variants of a DDM proposed, such as denoising diffusion probabilistic model (DDPM) [23, 15] and denoising diffusion implicit model (DDIM) [24], but we generalize these into DDMs with the common factors being a noising scheduling algorithm and a learned denoising process.

#### Forward Diffusion Process

The forward diffusion process gradually adds Gaussian noise to the data, leading to a series of latent variables $\mathbf{x}_1, \mathbf{x}_2, \ldots, \mathbf{x}_T$. The forward process is defined as:

$$q(\mathbf{x}_t \mid \mathbf{x}_{t-1}) = \mathcal{N}(\mathbf{x}_t \mid \sqrt{1 - \beta_t}\mathbf{x}_{t-1}, \beta_t \mathbf{I}), \tag{1}$$

where $\beta_t$ is the variance schedule that controls the amount of noise added at each step $t$.

Starting from the original data distribution $q(\mathbf{x}_0)$, the joint distribution over the sequence of latent variables is given by:

$$q(\mathbf{x}_{1:T} \mid \mathbf{x}_0) = \prod_{t=1}^{T} q(\mathbf{x}_t \mid \mathbf{x}_{t-1}), \tag{2}$$

and the marginal distribution of any $\mathbf{x}_t$ given $\mathbf{x}_0$ is:

$$q(\mathbf{x}_t \mid \mathbf{x}_0) = \mathcal{N}(\mathbf{x}_t \mid \sqrt{\bar{\alpha}_t}\mathbf{x}_0, (1 - \bar{\alpha}_t)\mathbf{I}), \tag{3}$$

where $\bar{\alpha}_t = \prod_{s=1}^{t}(1 - \beta_s)$.

#### Reverse Diffusion Process

The reverse diffusion process is modeled as:

$$p_\theta(\mathbf{x}_{t-1} \mid \mathbf{x}_t) = \mathcal{N}(\mathbf{x}_{t-1} \mid \boldsymbol{\mu}_\theta(\mathbf{x}_t, t), \sigma_t^2 \mathbf{I}), \tag{4}$$

where $\boldsymbol{\mu}_\theta(\mathbf{x}_t, t)$ is a neural network parameterized by $\theta$ that is trained to predict the mean, and $\sigma_t^2$ is typically set to $\beta_t$ or some other small variance.

The training objective is to minimize the variational lower bound on the negative log-likelihood:

$$L_{\text{vlb}} = \mathbb{E}_q \left[ D_{\text{KL}}(q(\mathbf{x}_T \mid \mathbf{x}_0) \| p(\mathbf{x}_T)) + \sum_{t=2}^{T} D_{\text{KL}}(q(\mathbf{x}_{t-1} \mid \mathbf{x}_t, \mathbf{x}_0) \| p_\theta(\mathbf{x}_{t-1} \mid \mathbf{x}_t)) - \log p_\theta(\mathbf{x}_0 \mid \mathbf{x}_1) \right], \tag{5}$$

where $D_{\text{KL}}$ denotes the Kullback-Leibler divergence, though the loss is often simplified to the mean squared error (MSE).

### 2.2 Quantum Theory for Machine Learning

#### Qubit States and Measurement

Quantum bits, or qubits, can exist in superpositions of states, allowing them to represent more information and process it more efficiently than classical bits [17]. Unlike a classical bit that can be either 0 or 1, a qubit can be in a superposition of both states simultaneously. Mathematically, the state of a qubit can be written as:

$$|\psi\rangle = \alpha|0\rangle + \beta|1\rangle, \tag{6}$$

where $|0\rangle$ and $|1\rangle$ are the basis states, and $\alpha$ and $\beta$ are complex numbers representing the probability amplitudes. These amplitudes are normalized so that:

$$|\alpha|^2 + |\beta|^2 = 1. \tag{7}$$

When a measurement is made on a qubit, the superposition collapses to one of the basis states. The probability of measuring the state $|0\rangle$ is $|\alpha|^2$, and the probability of measuring the state $|1\rangle$ is $|\beta|^2$.

One method of measurement relevant in quantum computing and QML is the Haar measurement. Haar measurement involves sampling unitary operations uniformly according to the Haar measure, which is the unique, invariant measure on the group of unitary matrices. This type of measurement is useful in QML because it provides a way to generate random quantum states and operations, which is important for algorithms that require randomization [13].

**Variational Quantum Circuits in Machine Learning**

Variational quantum circuits (VQCs) are a central tool in QML used in various architectures [7, 12, 20, 22, 9]. A VQC is a parameterized quantum circuit where some of the gates depend on adjustable parameters represented by quantum gates in the unitary operations. The general form of a VQC can be represented as:

$$|\psi(\boldsymbol{\theta})\rangle = U(\boldsymbol{\theta})|0\rangle, \tag{8}$$

where $U(\boldsymbol{\theta})$ is a unitary operation that depends on the parameters $\boldsymbol{\theta}$.

In the context of machine learning, these parameters are optimized using classical optimization techniques to minimize the cost function that measures the performance of the circuit throughout training. The power of VQCs in machine learning arises from their ability to exploit superposition and entanglement to potentially represent and solve problems more efficiently than classical algorithms [5].

## 2.3 Quark-Gluon Data Description

In this work, we generate quark and gluon jet data from the open-source LHC Compact Muon Solenoid (CMS) detector data. The data contains two classes that follow different distributions: hits from quark and gluon jets. Each sample is captured by three CMS subdetectors: electromagnetic calorimeter (ECAL), hadronic calorimeter (HCAL), and the reconstructed tracks as described in [1]. The dataset is preprocessed to crop a subset of 1,000 ECAL-detected jets of 125x125 pixels to 16x16 pixels for faster inference.

## 3 Related Work

Multiple works have combined the classical DDM and quantum algorithms. The paper in [28] proposed a fully quantum DDPM (QuDDPM) for generating an unknown quantum state distribution. Their method consists of applying random unitaries to quantum states, thus scrambling them into noise, and summing three error functions to optimize during training. The QuDDPM performed best compared to a quantum GAN and a quantum direct transport model.

Another work in [10] proposed a hybrid quantum-classical DDM. The algorithm involved a quantum denoising U-Net and classical noising and optimization. Good results were achieved on the MNIST dataset. However, tests on MNIST are not generalizable to other data, and a physics-conscious approach is required for the more complex Quark-Gluon data.

## 4 Methodology

In this section, we outline the methods used to construct fully quantum, hybrid, and fully classical models. The pipeline in Figure 1 shows all the models and their combinations.

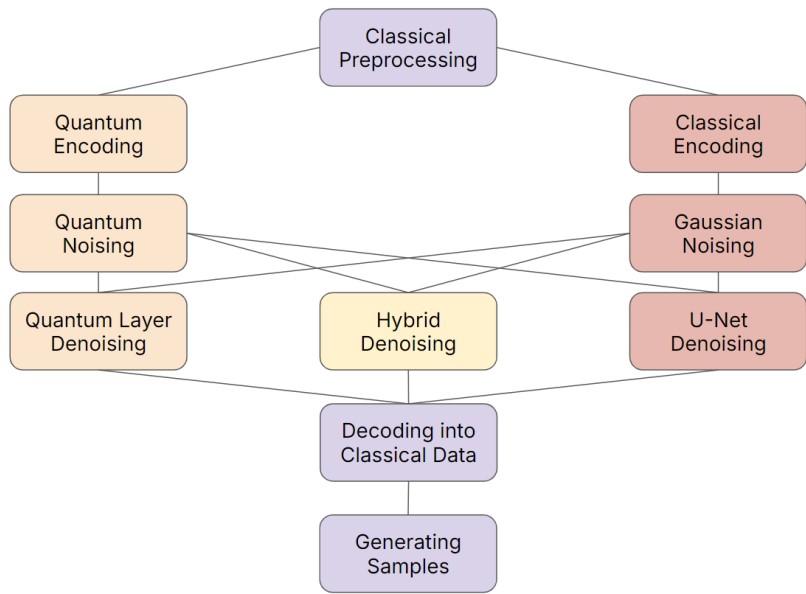

Figure 1: The pipeline that the data goes through with all the possible classical-quantum combinations of the forward and backward diffusion process.

## 4.1 Quantum Embedding

The first step of the pipeline is embedding classical data into quantum. We implemented angle encoding with $Rx$ rotation gates in groups of four pixels. Each sample is split into four channels as shown in Figure 2.

## 4.2 Forward Quantum Diffusion

The forward diffusion process was inspired by scrambling implementation in [28] using Haar random unitaries to mimic random noise application as shown in Figure 3. Since the choice of forward scrambling does not significantly impact model performance, as the authors in [3] have shown, an arbitrary noising transformation can be used. We chose to use the Haar measure for the quantum model as it allows for the unitary matrix transforms to scale with increasing resolution and dataset size. The final unitary is applied to each encoded channel, avoiding costly calculations associated with each timestep.

## 4.3 Denoising Quantum U-Net

Similar to classical DDMs, the denoising neural network is trained on learned parameters with the MSE loss function. The hybrid model uses a quantum strongly entangling layer surrounded by fully convolutional layers, and the fully quantum model relies only on the quantum layers. The circuit in the quantum layer, consisting of rotation and strongly entangling gates, is kept uniform across all models, with the number of layers treated as a tunable parameter.

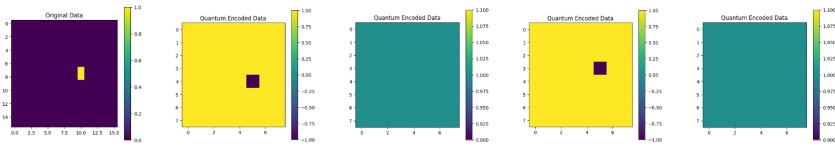

Figure 2: A sample of an encoded jet image.

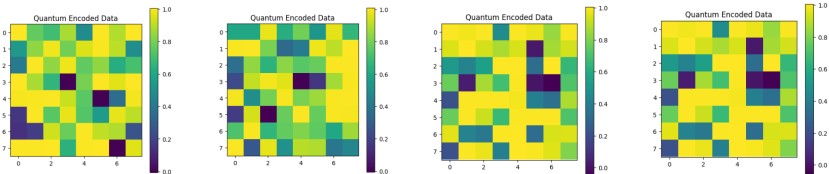

Figure 3: Haar noise applied to one encoded sample of four channels.

# 5    Experiments

In each experiment, we compare the performance using the Fréchet Inception Distance (FID) function, originally introduced in [14] for evaluating GANs, along with the loss function. The FID is defined as:

$$\mathrm{FID}(x, g) = \|\mu_x - \mu_g\|_2^2 + \mathrm{Tr}(\Sigma_x + \Sigma_g - 2(\Sigma_x \Sigma_g)^{1/2}),$$

where $\mu_x$ and $\mu_g$ are the mean feature vectors of the real and generated images, respectively, and $\Sigma_x$ and $\Sigma_g$ are their corresponding covariance matrices. Though we do not currently have access to quantum computers, we can simulate the behavior of quantum systems using simulators like Pennylane [4] and compare the results to classical models.

First, we present the loss and FID functions of the classical, hybrid, and quantum models, respectively, in Figure 4. Training on 50 epochs seems to be sufficient to reach convergence for all models.

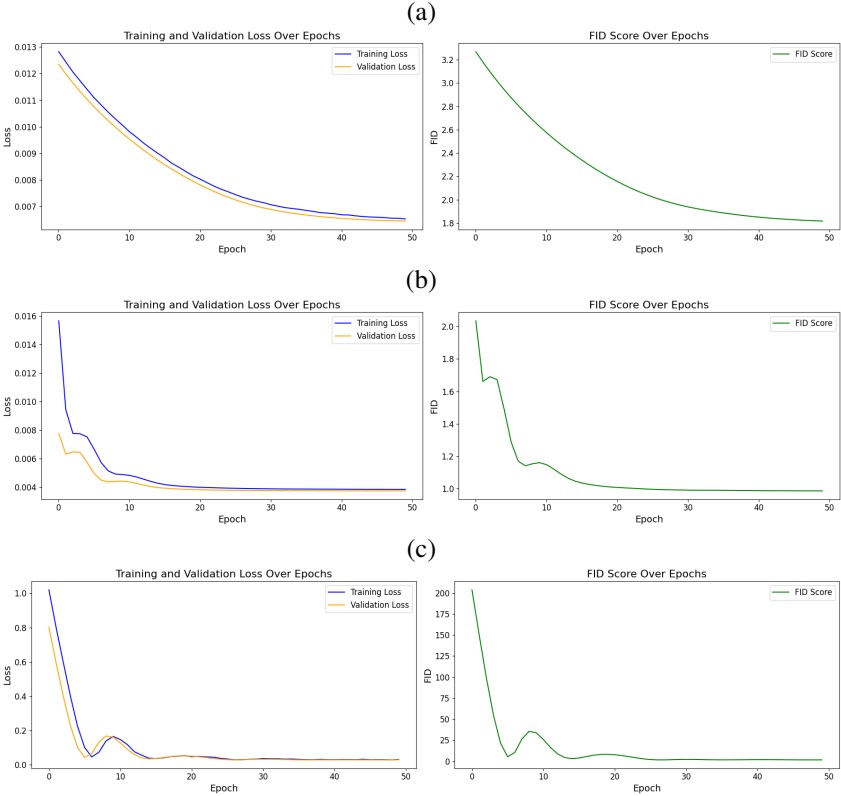

Figure 4: The losses and FID graphs of fully classical (a), hybrid (b), and fully quantum (c) models. For all models, MSE and Adam optimizer was used to reach convergence, and the FID function remained the same.

Some qualitative results in the form of generated samples are visualized in Figure 5. The generated images correspond to the best-performing model, which is the fully quantum architecture.

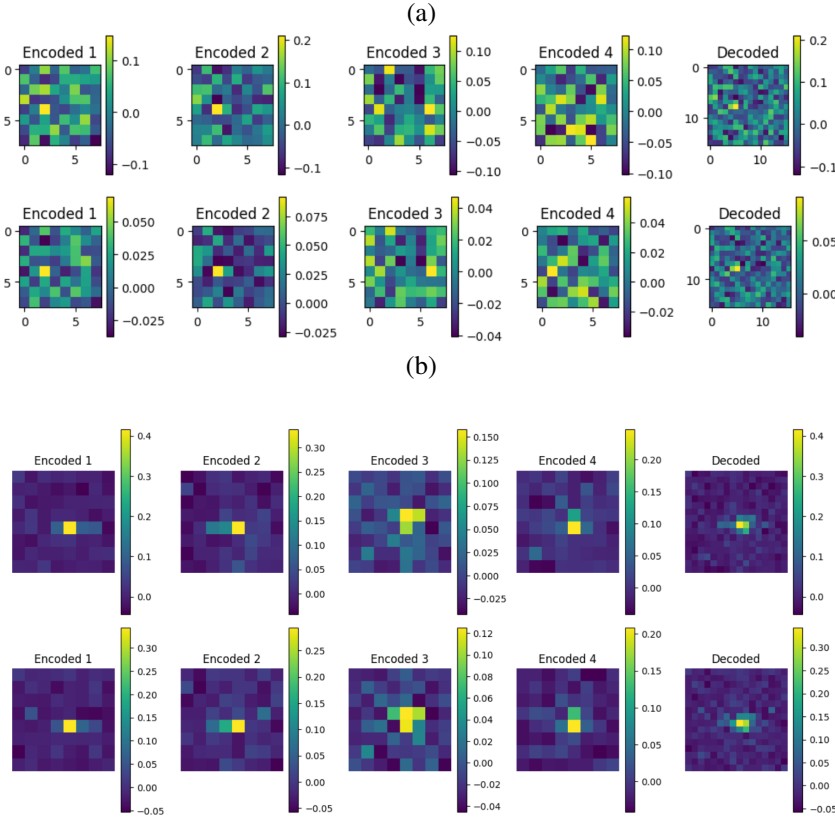

Figure 5: Samples generated from random noise with the four encoded channels on the left four rows and the decoded image on the far right for each sample. Subfigure (a) uses the hybrid model and (b) uses the fully quantum model.

## 6   Discussion

All models exhibited a significant decrease in loss, approaching near-zero values, which confirms effective learning. The FID scores followed a similar downward trajectory, with final values of 1.8169, 1.8123, and 2.7362. These results show that the performance of the models that leverage quantum circuits is comparable in relation to a structurally similar classical model. This implies that all or some parts of deep neural network computations can be offloaded to faster quantum processors to reduce training time and without a performance trade-off.

A potential reason for the plateau in FID scores across all models could be the sparsity of the data, where each sample contains only a few non-zero points. This sparsity may prevent the complete removal of noise in some of the encoded channels. A possible solution is a post-processing step where only the most prominent values are retained in the decoded data, increasing the confidence in jet locations. As a result, the generated samples would align more closely with the originals, as shown in Figure 6.

## 7   Conclusion and Future Direction

This work marks significant progress in leveraging quantum computing for machine learning applications, specifically for a generative DDM. In the future, our aim is to extend the models to generate all three jet channels for every sample. Additionally, experimenting with different parametric circuits and

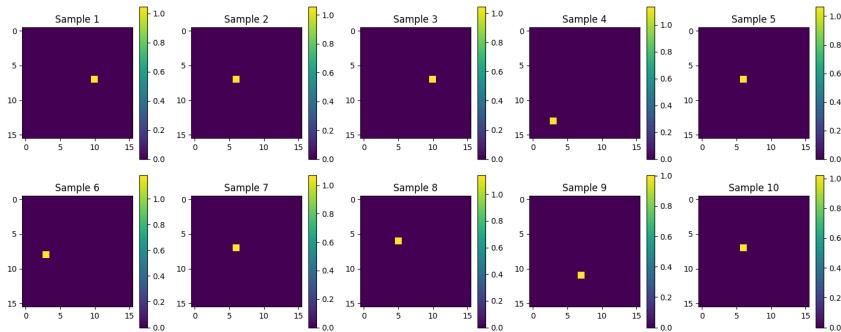

Figure 6: The most likely jet values of hybrid generated jets.

forward scrambling, such as using Gaussian transformations instead of Haar unitaries, may provide further insight into the impact that the architecture has on quantum generative learning. Finally, testing the quantum model on hardware with a limited number of qubits under practical constraints will be crucial in evaluating its scalability, real-world performance, and resilience to quantum noise.

# 8 Data Availability

The code is open source and can be found here: `https://github.com/mashathepotato/GSoC-Quantum-Diffusion-Model`.

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
