# OpenReview forum: "Quantum Diffusion Model for Quark and Gluon Jet Generation"
_NeurIPS.cc/2024/Workshop/MLNCP — MLNCP Poster_

### Official Review · Reviewer_8Avj · 2024-09-21
**Numerical experiments not fully specified; architecture seems to be unscalable**

**Rating:** 4
**Confidence:** 5

**Review:**

The authors propose using quantum circuits---potentially in conjunction with classical models---as a diffusion-based model for a generative modeling task relevant to high energy physics. This sort of problem---where the data is generated by a quantum process---is a good prospect for quantum machine learning to achieve a relevant advantage over classical machine learning models. The quantum circuit the authors consider to achieve this begins by scrambling the inputs using a Haar random circuit, followed by a "denoiser" of a trained quantum circuit.

However, the authors do not make any attempt to consider the scaling of their proposed implementation. Given the input to the denoising half of the quantum circuit is Haar random, from the arguments of, e.g., [arXiv:2312.09121](https://arxiv.org/abs/2312.09121) these networks will always have exponentially-vanishing gradients. This means variationally training such a network using a quantum computer would take time exponential in the problem size due to the exponential concentration of the loss landscape.

Furthermore, the authors' numerics are somewhat opaque; how many qubits were they performed using? The authors mention some form of channel-splitting procedure, but it is not clear what the final resource costs are of the network. More details on these numerics would be required for fully understanding how fair the comparison with the corresponding classical model is.

---

### Official Review · Reviewer_aXFD · 2024-10-04
**Very interesting approach of diffusion models introducing quantum computing techniques though practical implementations in hardware are not much clear.**

**Rating:** 7
**Confidence:** 3

**Review:**

Summary:
This paper introduces an innovative diffusion model leveraging quantum computing techniques for quark and gluon jet generation in high-energy physics. Their model integrates quantum random unitary matrices in the forward process and a variational quantum circuit within a U-Net architecture.  They are trying to address computational and resource challenges in generative modeling though its performance on real quantum hardware remains to be seen.

Strengths:
- The use of quantum circuits within diffusion is an exciting new approach for complex tasks like quark and gluon jet generation. This integration of quantum methods could significantly reduce computational resource demands compared to classical methods.
- They have introduced a clear and well-written methodology, from quantum embedding to the denoising process along with the underlying physics.
- Their simulations show promising results with performance metrics of FID score for both the hybrid and fully quantum models where quantum models perform slightly better than classical counterparts in terms of FID scores.


Weaknesses:
- They have used quantum simulators like Pennylane, but do not test the model on actual quantum hardware. While the results from simulators are encouraging, the real-world performance and scalability of the model under practical quantum constraints are not clear.
- Their dataset is with reduced dimensions (from 125x125 to 16x16 pixels), but it remains unclear how the model would scale to larger datasets or more complex structures. Further exploration could enhance the clarity.
- Do you think there is any overhead in managing quantum circuits and gates that may offset some of the computational gains unless quantum hardware improves significantly?
-The figure quality could be improved significantly, and the FID scores from all three methods would be easier to compare. Reference to the figure is missing on page 4 having a "question mark".

---

### Decision · Program_Chairs · 2024-10-10

Accept (Poster)